# Armed violent conflict and healthcare-seeking behavior for maternal and child health in sub-Saharan Africa: A systematic review

Gbadebo Collins Adeyanju[1,2,3]*, Pia Schrage[4], Rabiu Ibrahim Jalo[5,6], Liliana Abreu[7], Max Schaub[8,9]

**1** Media and Communication Science, University of Erfurt, Erfurt, Germany, **2** Centre for Empirical Research in Economics and Behavioural Science (CEREB), University of Erfurt, Erfurt, Germany, **3** Psychology and Infectious Diseases Lab (PIDI), University of Erfurt, Erfurt, Germany, **4** Willy Brandt School of Public Policy, University of Erfurt, Erfurt, Germany, **5** Department of Community Medicine, Faculty of Clinical Sciences, Bayero University, Kano, Kano State, Nigeria, **6** Department of Community Medicine, Aminu Kano Teaching Hospital, Kano, Kano State, Nigeria, **7** Department of Politics and Public Administration, University of Konstanz, Konstanz, Germany, **8** Department of Political Science, University of Hamburg, Hamburg, Germany, **9** WZB Berlin Social Science Center, Berlin, Germany

* gbadebo.adeyanju@uni-erfurt.de

## Abstract

### Background

Over 630 million women and children worldwide have been displaced by conflict or live dangerously close to conflict zones. While the adverse effects of physical destruction on healthcare delivery are relatively well understood, the effects on healthcare-seeking behavior remain underexplored, particularly in sub-Saharan Africa. This study aims to better understand the interconnections and knowledge gaps between exposure to armed violent conflicts and healthcare-seeking behaviors for maternal and child health in sub-Saharan Africa.

### Methods

Five key electronic databases (PubMed, Scopus, Web of Science, PsycNET, and African Journals Online) were searched for peer-reviewed publications between 2000 and 2022. The review was designed according to PRISMA-P statement and the protocol was registered with PROSPERO database. The methodological quality and risks of bias were appraised using GRADE. A data extraction instrument was modelled along the Cochrane Handbook for Systematic Reviews and the Centre for Reviews and Dissemination of Systematic Reviews.

### Result

The search results yielded 1,148 publications. Only twenty-one studies met the eligibility criteria, reporting healthcare-seeking behaviors for maternal and child health. Of the twenty-one studies, seventeen (81.0%) reported maternal health behaviors such as antenatal care, skilled birth attendance, postnatal care services, and family planning. Nine studies (42.8%) observed behaviors for child health such as vaccination uptake, case management for

**Data Availability Statement:** The datasets used and/or analyzed during the current study are available on https://osf.io/gnkyc/.

**Funding:** University of Hamburg provided funding as part of research support to its faculty members. The funders had no role in study design, data collection and analysis, decision to publish, or preparation of the manuscript.

**Competing interests:** The authors have declared that no competing interests exist.

**Abbreviations:** AJOL, African Journals Online; ANC, Antenatal Care; BCG, Bacillus Calmette–Guérin vaccine; BH, Boko Haram; CPR, Contraceptive Prevalence Rate; CRD, Centre for Reviews and Dissemination of Systematic Reviews; DPT, Diphtheria-Pertussis-Tetanus vaccine; DRC, Democratic Republic of Congo; EIB, Early Initiation of Breastfeeding; FP, Family Planning; GRADE, Grades of Recommendation, Assessment, Development and Evaluation; ICRC, The International Committee of the Red Cross; IDP, Internally Displaced Person; LGA, Local Government Area; MCH, Maternal and Child Health; MeSH, Medical Subject Headings; NGO, Non-Governmental Health Organization; OPV, Oral Polio Vaccine; PICO, Population, Intervention, Comparison and Outcomes; PNC, Postnatal Care; PRISMA-P, The Preferred Reporting Items for Systematic Reviews & Meta-Analyses Protocols; SBA, Skilled Birth Attendant; SSA, Sub-Saharan Africa; UCDP, The Uppsala Conflict Data Program; UNDP, United Nations Development Programme; WHO, World Health Organization.

pneumonia, diarrhea, malnutrition, and cough. While conflict exposure is generally associated with less favorable healthcare-seeking behaviors, some of the studies found improved health outcomes. Marital status, male partner attitudes, education, income and poverty levels were associated with healthcare-seeking behavior.

## Conclusion

There is a need for multifaceted interventions to mitigate the impact of armed violent conflict on healthcare-seeking behavior, given its overall negative effects on child and maternal healthcare utilization. While armed violent conflict disproportionately affects children's health compared to maternal health, it is noteworthy that exposure to such conflicts may inadvertently also lead to positive outcomes.

## Prospero registration number

CRD42023484004.

## Introduction

The health status of populations, especially women and children, tends to deteriorate in countries affected by armed conflict. Exposure to armed conflict often leads to an overwhelming loss of human life, disruptions of health service delivery, infrastructure breakdown and widespread displacement [1]. A substantial portion of the world's women and children resides in countries suffering from armed conflicts [2]. Estimates indicate that, in 2017, approximately 630 million women and children, accounting for 10% of women and 16% of children worldwide, either faced displacement due to conflict or resided dangerously close to conflicts zones [2–5]. The problem is particularly pronounced in sub-Saharan Africa (SSA), where armed conflicts have been more prevalent than in other regions. For example, nearly 70% of all countries in SSA have witnessed armed conflicts since 1980 [2, 6].

The International Committee of the Red Cross (ICRC) classifies armed conflicts as non-international conflicts involving governmental and non-government forces, including insurgency, organized crime and banditry [7]. Meanwhile, the Uppsala Conflict Data Program (UCDP) sees armed conflict as a contested incompatibility related to government and/or territory, wherein the use of armed force between the military forces of two parties, with at least one being a government of a state, has resulted in at least 25 battle-related deaths each year [8]. It is widely recognized that the impact of wars extends far beyond the cease-fire [9–11]. However, the underlying reasons for this phenomenon are yet to be fully understood. Some adverse long-term effects can be directly attributed to the conflict itself: unexploded ordnance continues to pose a threat to lives and livelihoods even decades after a war [12], and injuries sustained during conflicts shorten lives and compromises the health of those affected [13].

Wars have health consequences beyond the physical injuries caused by violence: destruction of sanitation and energy infrastructure can lead to the spread of waterborne and respiratory diseases [14]. Reduced functioning or restricted access to health facilities may reduce vector control and contribute to the spread of mosquito-transmitted diseases such as malaria or dengue fever [15, 16]. Refugees and internally displaced persons may inadvertently introduce diseases into other areas [17, 18]. The destruction of hospitals and clinics, combined with the departure of health workers seeking safety for themselves and their families, further

exacerbates the health challenges in conflict zones [19–23]. Other factors that have been associated with maternal health outcome in armed violent conflicts include miscarriages, premature deliveries, unsafe living conditions and malnutrition [24].

Beyond tangible impacts, armed violent conflicts can have intangible effects, including the normalization of violence, shifts in social norms, and the breakdown of the social order. These elements are often accompanied by increased levels of sexual violence and voluntary unprotected sex, leading to the spread of sexually transmitted diseases [25]. The trauma resulting from violence, even if not visibly apparent, is associated with both persistent mental health issues and associated physical ailments that often afflict victims throughout their lives [11]. These latter effects may cause maternal behavioral changes such as smoking or excessive alcohol consumption/addiction, which, in turn, can negatively affect child health in a variety of ways, from pregnancy to adverse effects on birth weight, child height for age, child weight for height, immunization coverage, infant and child mortality [26–32]. Moreover, exposure to violence may lead to fear, mistrust, and psychological trauma, creating reluctance among individuals to seek out health-services [33–35]. However, while the consequences of structural armed violent conflicts abound, there is a dearth of evidence on the direct relationship between health-seeking behavior on account of experience of or exposure to armed violent conflicts in the SSA.

Hence, this study aims to generate new insights into the complex interplay between health decision-making or health-seeking behavior and exposure to armed violence during and after conflicts. It seeks to generate knowledge that can be used in health promotion initiatives in conflict and post-conflict societies, especially those that aim at reducing child and maternal mortality and morbidity. The specific objectives of this study are to assess the existing body of knowledge on healthcare-seeking behaviors in armed violent conflict settings, and to explore and describe the various factors influencing healthcare-seeking behaviors in regions of SSA affected by armed violent conflicts.

## Methods

### Systematic review registration and reporting

The review protocol was registered in the PROSPERO database (Reg. no.: CRD42023484004) and designed in accordance with the Preferred Reporting Items for Systematic Reviews & Meta-Analyses Protocol (PRISMA) 2020 Statement [36]. As this is a systematic review of previously published studies, ethical review and informed consent are not applicable.

### Data sources and search strategy

An interdisciplinary review team with expertise in global health, public health, public policy, epidemiology, political science and conflict studies, social science and evidence synthesis was constituted. Guided by the Population, Intervention, Comparison and Outcomes (PICO) model for systematic review [37, 38] (see Table 1), the team developed a broad research question: *"How does exposure to armed violent conflicts affect healthcare-seeking behaviors for maternal and child health in sub-Saharan Africa?"*.

A systematic search was conducted on four general and one key electronic database. The general databases were: MEDLINE via PubMed Central, Scopus, Web of Science and PsycNET. Because of its rich repository of Africa-based journal publications, we also included the African Journals Online (AJOL). Suitable MeSH and thesaurus terms were adapted with key terms in each database searched. Three strings of multiple combinations of search terms and relevant Boolean operators to the study PICO framework were developed as seen in Table 2, such as Population (parents and caregivers of under-5, children under-5, pregnant women,

**Table 1. The PICO framework for systematic review.**

| PICO | Measure | |
|---|---|---|
| **Population** | Parents or caregivers of under-fives, pregnant women, and women in SSA | Children, child, maternal, prenatal, antenatal, pregnancy, all countries in SSA. |
| **Intervention or exposure** | Experience of (exposure to) armed violent conflicts | War, armed conflict, conflict experiences, conflict exposure, organized violence, armed violence, fighting, insurgency, war trauma, traumatic experiences, civil conflict, conflict-affected populations, war-affected communities |
| **Comparator** | Not Applicable | Not Applicable |
| **Outcome** | Healthcare-seeking behavior | Healthcare-seeking behaviour; healthcare-seeking behavior; health service utilization; health service utilization; healthcare utilization; healthcare utilisation; healthcare access; access to healthcare; health behavior; health behaviour; prenatal care; postnatal care; healthcare decision-making, healthcare choices, healthcare barriers, delayed care; refused healthcare; healthcare; seeking; behaviour; behavior; health; service; utilization; utilisation; use of maternal health services, use; maternal; services; healthcare access; access to healthcare; utilization of healthcare services; utilisation of healthcare services |

**Table 2. Search strategy: #1, #2, and #3.**

| Search string #1 | Search string #2 | Search string #3 |
|---|---|---|
| "armed conflict" OR "armed violence" OR "insurgency" OR "conflict experiences" OR "conflict exposure" OR "war trauma" OR "traumatic experiences" OR "civil conflict" OR "conflict-affected populations" OR "war-affected communities" AND | "maternal health" OR "child health" OR "reproductive health services" OR "childhood health" OR "maternal health services" OR "healthcare-seeking behavior" OR "health service utilization" OR "healthcare" OR "seeking" OR "behaviour" OR "behavior" OR "health" OR "service" OR "utilization" OR "use of maternal health services" OR "use" OR "maternal" OR "services" OR "healthcare access" OR "access to healthcare" OR "healthcare utilization" OR "utilization of healthcare services" OR "healthcare decision-making" OR "healthcare choices" OR "healthcare barriers" AND | "Africa"[MeSH] OR Africa*[tw] OR Algeria[tw] OR Angola[tw] OR Benin[tw] OR Botswana[tw] OR "Burkina Faso"[tw] OR Burundi[tw] OR Cameroon [tw] OR "Cape Verde"[tw] OR "Central African Republic"[tw] OR Chad[tw] OR Comoros[tw] OR Congo[tw] OR "Democratic Republic of Congo"[tw] OR Djibouti[tw] OR "Equatorial Guinea"[tw] OR Eritrea[tw] OR Ethiopia[tw] OR Gabon[tw] OR Gambia[tw] OR Ghana[tw] OR Guinea[tw] OR "Guinea Bissau"[tw] OR "Ivory Coast"[tw] OR "Cote d'Ivoire"[tw] OR Kenya[tw] OR Lesotho[tw] OR Liberia[tw] OR Madagascar[tw] OR Malawi[tw] OR Mali[tw] OR Mauritania[tw] OR Mauritius[tw] OR Mayotte[tw] OR Mozambique[tw] OR Mocambique [tw] OR Namibia[tw] OR Niger[tw] OR Nigeria[tw] OR Principe[tw] OR Reunion[tw] OR Rwanda[tw] OR "Sao Tome"[tw] OR Senegal[tw] OR Seychelles[tw] OR "Sierra Leone"[tw] OR Somalia[tw] OR "South Africa"[tw] OR "St Helena"[tw] OR Sudan[tw] OR Swaziland[tw] OR Tanzania[tw] OR Togo[tw] OR Uganda[tw] OR "Western Sahara"[tw] OR Zaire[tw] OR Zambia[tw] OR Zimbabwe[tw] OR "Central Africa"[tw] OR "Central African"[tw] OR "West Africa"[tw] OR "West African"[tw] OR "Western Africa"[tw] OR "Western African"[tw] OR "East Africa"[tw] OR "East African"[tw] OR "Eastern Africa"[tw] OR "Eastern African"[tw] OR "South African"[tw] OR "Southern Africa"[tw] OR "Southern African"[tw] OR "sub Saharan Africa"[tw] OR "sub Saharan African"[tw] OR "subSaharan Africa"[tw] OR "subSaharan African"[tw] OR Africa South of the Sahara[MH:EXP] NOT ("guinea pig"[tw] OR "guinea pigs"[tw] OR "aspergillus niger"[tw] |

and mothers in SSA), Intervention (experience of or exposure to armed violent conflicts), Comparison (none) and Outcome (healthcare-seeking behavior: e.g., maternal and child health). Two independent reviewers conducted the initial screening of the titles, abstracts, and keywords of relevant studies, prior to the review of the full text involving all authors.

## Study selection process

The search results were imported into the Zotero reference management software to compile and filter relevant articles that meet the eligibility criteria and to exclude duplicates [39]. In the initial stage, five authors (GCA, LA, MS, PS, and RIJ) independently reviewed all articles using titles, keywords, and abstracts. This was done to determine the suitability of the article for the purpose of the review and based on defined inclusion and exclusion criteria. In the second stage, full-text articles of all included studies were scanned to determine final eligibility. All included studies from the first and second stages were proportionally distributed and assessed strictly against the inclusion and exclusion criteria described in Table 3. Disagreements or non-aligned decisions were resolved through weekly workshops with all authors. The PRISMA flow chart for systematic review was used to document the study selection process (see Fig 2).

## Study eligibility

Studies were included in the review according to the PICO criteria. Predefined inclusion and exclusion criteria were developed to avoid bias due to non-relevant studies (see Table 3). Due to the sensitivity of the topic and to avoid an irregular time frame (using month instead of year of publication), i.e., 2000 –October 2023, the study used articles published within the year 2000–2022. Also, using articles of this sensitive nature without adequate time for feedback from the scientific community and the public could be risky due to potential corrections and retrievals.

## Outcome of interest

Since studies on healthcare-seeking behaviors encompass a diverse demographic spectrum, ranging from infants to the elderly, this study targeted parents/legal guardians (caregivers) of children under the age of five years, pregnant women, and mothers in SSA. Healthcare-seeking behavior has been defined as any action taken by individuals who perceive themselves to have a health problem or are ill, for the purpose of finding an appropriate remedy [40, 41]. Given the broad nature of healthcare-seeking behavior, this review focuses specifically on the utilization of health services in armed conflict settings.

**Table 3. Inclusion and exclusion criteria.**

| Inclusion | Exclusion |
|---|---|
| a. English language only. | a. Published research articles in non-peer-reviewed journals. |
| b. Only publications between 2000–2022 | |
| c. All countries within the sub-Saharan Africa region only, as defined by United Nations development Program [38]. | b. Studies focused on gender-based or domestic violence. |
| d. All studies that assess experience or exposure to armed violent conflicts and its consequent impact on health outcomes. | c. Studies whose primary or secondary outcomes are not healthcare-seeking behavior or health decision-making. |
| e. All studies with outcomes significantly impacting behavior and decision-making, leading to changes in health. | d. Studies on the cost of healthcare in armed conflict settings. |
| f. Only studies which assess maternal and child health. | e. Studies that report findings only on health behavior or health changes not influenced by armed violent conflict. |
| | f. Published research articles in 2023. |

The impacts of exposure to or experience of armed violence conflict on health-seeking behavior in SSA were classified into two broad categories. Maternal health variables include antenatal care attendance, maternal vaccination, skilled attendance at birth, and postnatal care services (e.g., utilization of family planning services). Child health variables include the uptake of childhood vaccination from routine immunization systems, and healthcare-seeking behaviors for common health problems among children under the age of five years, such as fever and diarrhea.

## Quality assessment of studies and risk of bias

The quality of evidence in each study in the review was assessed using the Grades of Recommendation, Assessment, Development, and Evaluation Working Group [42]. The review adhered to the World Health Organization (WHO) and Cochrane Collaboration on the principles of the GRADE system for evaluating the quality of evidence for outcomes reported in systematic reviews [43]. The authors independently assessed each relevant study using the criteria from the Cochrane Collaboration and the Centre for Reviews and Dissemination (CRD) to evaluate the risk of bias and quality of evidence [44].

## Interrater reliability

To assess inter-rater reliability, 10 studies were randomly selected from the 1,148 articles initially identified. Each study was independently assessed using the predefined eligibility criteria, and if there was disagreement about the decision, the process was repeated. Inconsistent decisions were discussed as a group to ensure agreement among the authors. The resulting interrater reliability Kappa coefficient was K = 0.95 (where 1.0 is a perfect score), which justified the approach of the subsequent screening processes [45].

## Data extraction

A data extraction instrument was developed using Microsoft Excel, based on the Cochrane Handbook for Systematic Reviews and the CRD's Guidance for Undertaking Reviews in Health Care [44, 46]. Two algorithms were formulated. The first concerned the characteristics of the study results. It comprised key indicators such as author last name, publication year, title of publication, country of study, study design, primary and secondary outcomes, nature of armed violence conflict, study setting, gender of the targeted group, and factors associated with health-seeking behavior that were reported in the studies, especially those related to maternal and child health. The second algorithm was used to assess the quality of evidence and risk of potential biases in each study, such as study design, selection, detection, reporting, attrition, and publication bias. In addition, the assessment considered evidence related to impression, inconsistency, and indirectness in the studies. The factors influencing health-seeking behavior due to exposure to armed violent conflict identified in the studies were systematically categorized into themes. Two authors (PS and RIJ) independently extracted data from the final included studies, and populated the two designed matrices, which were then peer-reviewed in a workshop among all authors to confirm the collected data and resolve disagreements or misinterpretations.

## Study analysis

Significant evidence relevant to the goal of the review was extracted, appraised and reported in a systematic fashion using the PRISMA-P checklist [47, 48]. The extracted data were synthesized to answer the research question.

## Results

Fig 1 visualizes the country-contexts of the final set of studies that met all the review criteria.

The included studies cover nine countries: Nigeria, Burkina Faso, DRC, Sudan, South Sudan, Somalia, Uganda, Kenya, and Burundi: All nine countries have experienced prolonged armed violent conflicts over several decades. In Somalia, since the formation of the Al Qaeda-affiliated Al-Shabaab group, at least 1,000 deaths have been reported every year [49]. Widespread conflict has resulted in the deterioration of Somalia's public health system with over 2.6 million internally displaced persons due to the conflict and drought [49, 50]. In eastern DRC, nearly two decades of armed conflict have severely compromised the health system, contributing to an estimated 3.9 million excess deaths [51, 52]. Uganda is currently recovering from over 20 years of armed conflict orchestrated by the Lord's Resistance Army, which resulted in

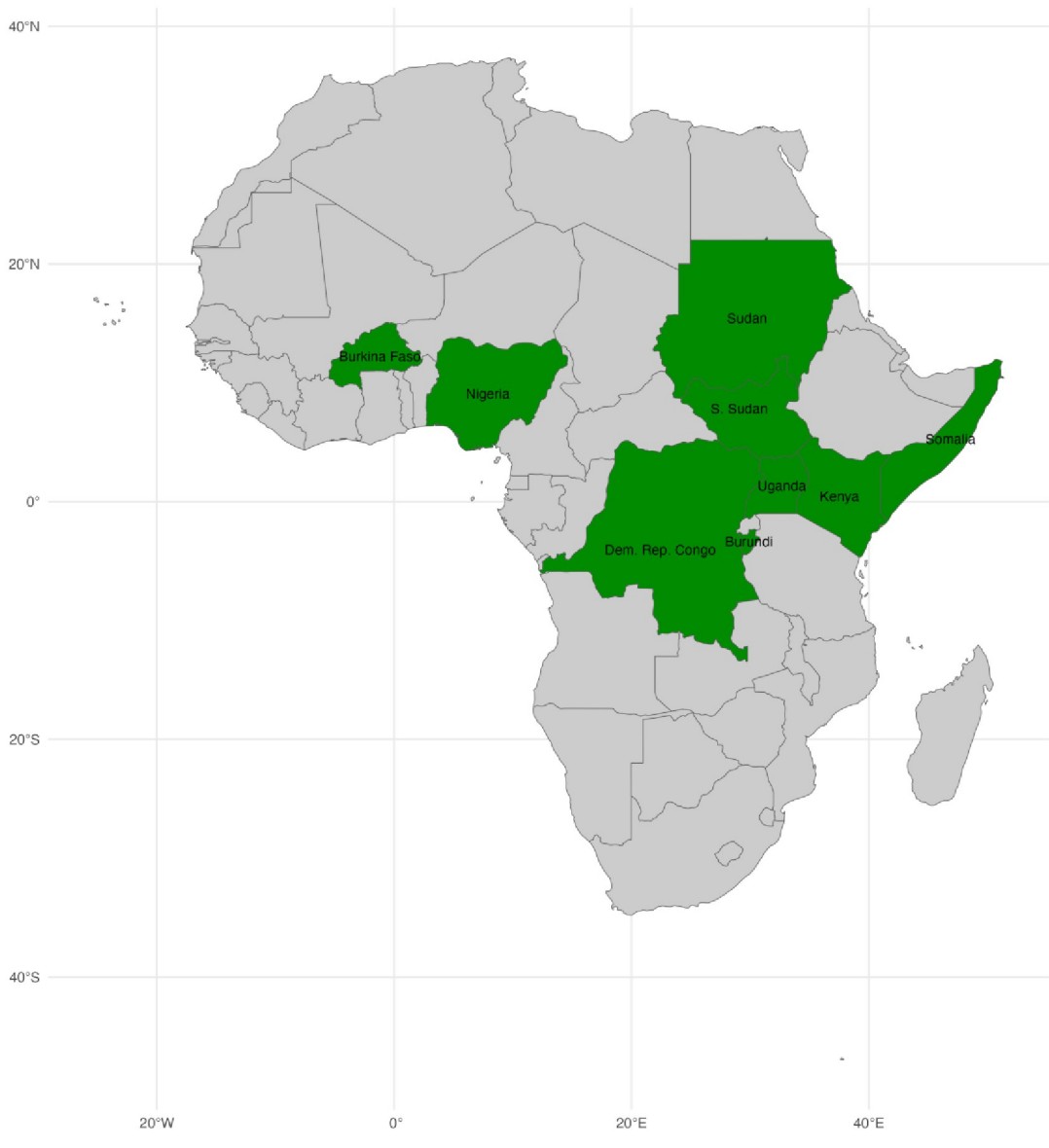

**Fig 1. Eligible studies by country across SSA.**

the disruption of health services and massive population displacement estimated at 2 million [53, 54]. Both Burundi and Uganda are recovering from atrocious civil wars that each claimed tens of thousands of lives and caused millions of people to be displaced [55]. In West African countries like Nigeria and Burkina Faso, constant conflicts due to religion, ethnicity, and political views have posed significant challenges [53]. Currently, Boko Haram operating in Nigeria is recognized as one of the most extreme terrorist groups in SSA. Since 2009, the Boko Haram insurgency has killed more than 20,000 people and displaced more than 2 million [56].

Nigeria, Somalia, the DRC, Burkina Faso, Sudan, South Sudan, and Cameroon have often experienced very high-intensity armed conflicts, while countries such as Kenya and Uganda have experienced lower-intensity conflicts. These conflicts have involved non-state actors, terrorists, bandits, and other armed groups. The complex humanitarian emergencies caused by violent armed conflict severely impact on the already fragile national and subnational health systems in many of these countries through the destruction of health facilities and the flight of trained health workers [57, 58].

## Search results

The search across five databases [PubMed, Scopus, Web of Science, PsycNET, and African Journals Online (AJOL)] yielded 1,148 articles (PubMed = 596, Scopus = 218, Web of Science = 137, PsycNET = 159 and AJOL = 38), with 77 articles removed as duplicates. Therefore, 1,071 articles were then screened for eligibility based on title and abstract. A total of 1,005 articles were excluded during this screening phase for various reasons: articles not in English (n = 3), not peer-reviewed (n = 17), health-seeking behavior not the primary outcome (n = 627), not focusing on armed conflict (n = 30), not studies in SSA (n = 86), studies of sexual or domestic violence (n = 91), studies of healthcare costs (n = 10), studies on internally displaced persons or refugees (n = 44), review articles (n = 36), and finally, studies whose focus was not maternal and child health (n = 61).

The remaining 66 articles underwent a full text screening. After the screening, 45 articles were excluded for the following reasons: duplicates (n = 2), non-peer-reviewed articles (n = 1), review articles (n = 8), outcome not defined (n = 2), health-seeking behavior not the primary outcome (n = 20), not focused on armed conflict (n = 3) and maternal and child health (n = 9), respectively. The final review incorporated 21 studies reporting healthcare-seeking behaviors for maternal and child health in conflict and post-conflict settings in SSA. The screening and selection process is illustrated in the PRISMA-P flow diagram of Fig 2.

A significant proportion of the eligible studies were from Nigeria (n = 8, 38.1%), Uganda (n = 3, 14.2%), and Congo (n = 3, 14.2%). All studies were peer-reviewed journal articles. Out of the total, 13 studies (61.8%) were descriptive in design, while three (14.3%) used a pre-post survey approach. 12 studies focused exclusively on maternal health issues (57.2%) and four (19.0%) involved children exclusively, while five studies reported outcomes for both children and women (23.8%). The majority of the studies (90.5%) were published within the last ten years (2012–2022), and none were randomized control trials. The characteristics of the included studies are shown in Table 4. Missing data in the eligible studies were very minimal, hence pose no relevance to the review outcomes. Studies whose missing data are above 0.5% are excluded.

As outlined in Table 5, the included studies met all eligibility criteria and reported different outcomes for healthcare-seeking behaviors in SSA. Of the twenty-one included studies, seventeen (81.0%) reported healthcare-seeking behaviors for maternal health services in settings of armed violent conflict. The reported healthcare-seeking behaviors were antenatal care attendance, skilled birth attendance, postnatal care services, family planning uptake/contraceptive prevalence rate, health facility births, caesarean section rates, unmet need for family planning,

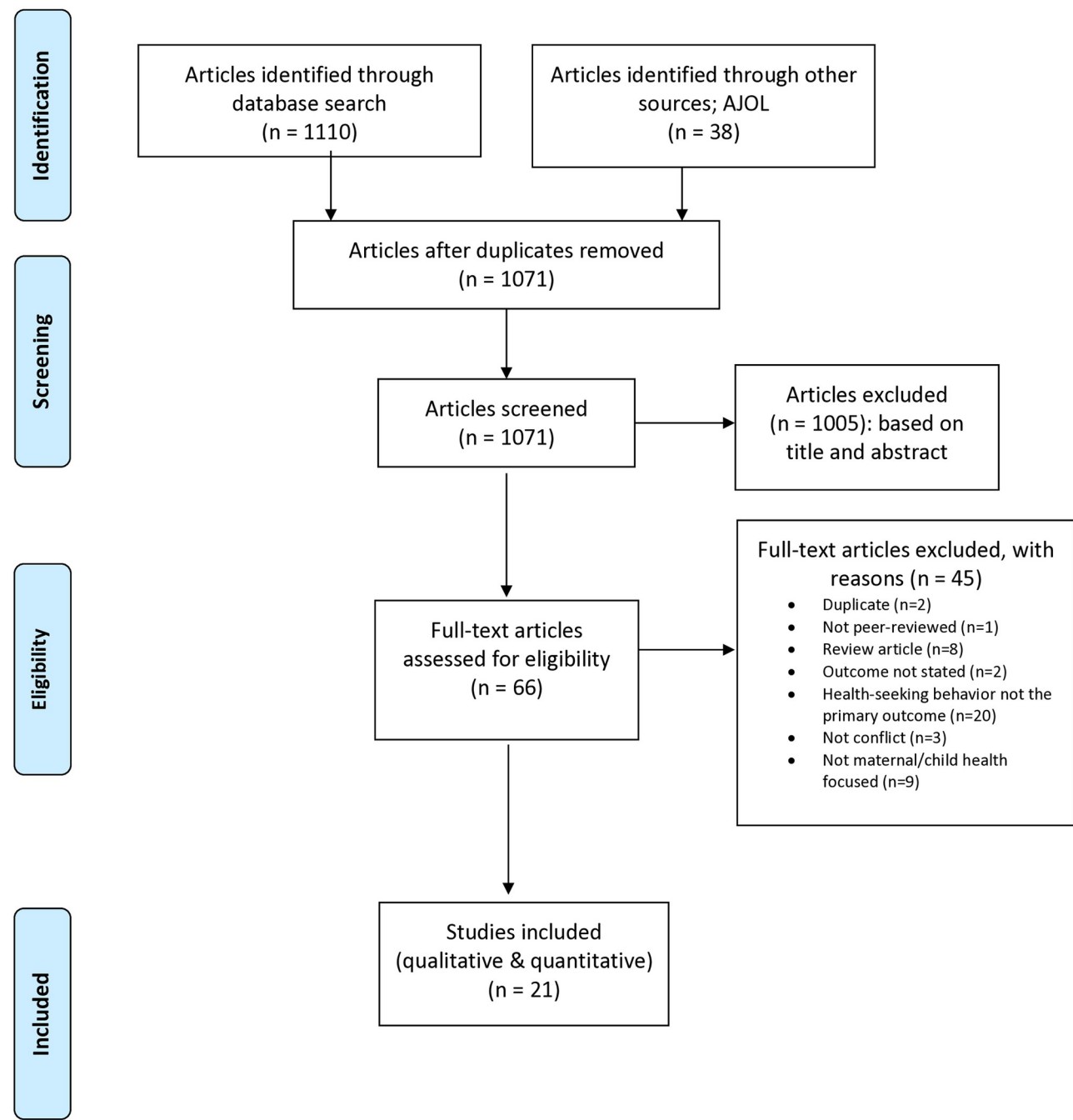

**Fig 2. PRISMA 2009 flow diagram [59].**

and HIV/AIDS care and treatment [19, 20, 49, 51, 55, 57, 60–70]. Nine studies (42.9%) reported healthcare-seeking behaviors for child health. The child health seeking behaviors reported were vaccines uptake (measles, BCG, and DPT), and case management for pneumonia, diarrhea, malnutrition, fever, and cough [49, 53, 60, 66, 68, 69, 71–73]. Antenatal care attendance was the most common healthcare behavior observed in most of the studies (57.1%), followed by utilization of skilled birth attendants (33.3%) for maternal health, while

**Table 4. Characteristics of included studies.**

| Variable | Frequency | Percentage (%) |
|---|---|---|
| **Year of study** | | |
| Before 2012 | 2 | 9.5 |
| 2012 and after | 19 | 90.5 |
| **Study design** | | |
| Cross sectional | 13 | 61.8 |
| Pre–post | 3 | 14.3 |
| Quasi-experimental | 1 | 4.8 |
| Longitudinal | 1 | 4.8 |
| Others | 3 | 14.3 |
| **Study setting** | | |
| Single country | 18 | 85.7 |
| Multi-country | 3 | 14.3 |
| **Type of health seeking behavior** | | |
| Maternal | 12 | 57.2 |
| Child | 4 | 19.0 |
| Maternal & child | 5 | 23.8 |
| **Nature of violent conflict experienced** | | |
| Armed conflict | 7 | 33.3 |
| Civil conflict | 5 | 23.8 |
| Insurgency | 9 | 42.9 |
| **Country of study** | | |
| Nigeria | 8 | 38.1 |
| Somalia | 1 | 4.8 |
| South Sudan | 1 | 4.8 |
| Uganda | 3 | 14.2 |
| Congo (DRC) | 3 | 14.2 |
| Burundi & Uganda | 2 | 9.5 |
| Burkina Faso | 1 | 4.8 |
| Sudan, Uganda & DRC | 1 | 4.8 |
| Kenya | 1 | 4.8 |
| **Study method** | | |
| Quantitative | 13 | 61.9 |
| Qualitative | 3 | 14.3 |
| Mixed methods | 5 | 23.8 |

the uptake of measles vaccination was the most common health-seeking behavior for child health.

The included studies reported a variety of outcomes for healthcare-seeking behaviors, with no overall consistent effect detectable. Ten studies reported a significant decrease in the use of health services for both maternal and child health, three reported an increase in the use of various health services, and four others reported mixed outcomes (increased use of maternal health services and decreased use in some areas). Healthcare Factors influencing healthcare-seeking behavior in armed conflict settings included gender, marital status, the influx of humanitarian aid, the nature and intensity of the armed conflict, distance from conflict areas, as well as individual, community, and contextual factors. Table 5 provides a key summary and synopsis of the findings from all the articles reviewed.

**Table 5. Summary of study findings.**

| No | Author | Country | Study Design | Population | Types of HSB | Types of maternal health | Types of child health | Nature of armed conflicts | Result | Result Effects | Primary Outcomes | Name of extractors and dates |
|---|---|---|---|---|---|---|---|---|---|---|---|---|
| 1 | Ahmed et al. 2020 | Somalia | Cross-sectional | Women and children (Under 5) | Maternal and child | ANC, CPR & Caesarean section rate | DPT, measles and case management for pneumonia | Armed conflict | Increase of most intervention coverage in Somali land; ANC decrease in Puntland | Conflict decreased ANC attendance in Puntland but not in Somali land | More than one outcome | GCA, RIJ & PS. 20.11.2023 |
| 2 | Babakura et al. 2021 | Nigeria | Other (retrospective) | Children (Under 5) | Child | NA | Measles vaccine uptake | Insurgency | Measles vaccination of 68% of targeted children | Higher coverage in LGAs without security challenges | Intervention effective | RIJ & PS. 20.11.2023 |
| 3 | Bayo et al. 2021 | South Sudan | Cross-sectional | Pregnant women and neonates | Maternal and child | ANC, health facility birth | Neonatal complications | Civil conflict | ANC visits declined by 21%; health facility births declined by 6.9% | Significant decline in key maternal and neonatal health service utilization indicators in Torit Hospital during conflict exposure | Uptake declined | GCA, RIJ & PS. 24.11.2023 |
| 4 | Casey et al. 2013 | Uganda | Pre-Post | Women | Maternal | FP use and unmet need for FP | NA | Armed conflict | Increase in use of modern FP methods; increase in use of long-acting and permanent methods; decrease in % of women with unmet need for FP | Women will make use of FP services in conflict settings when provided | Uptake increased | GCA, RIJ & PS. 24.11.2023 |
| 5 | Casey et al. 2017 | DRC | Pre-Post | Women | Maternal | FP | NA | Armed conflict | Increase in use of modern FP methods and long acting and permanent methods | Demand for FP methods, is present in humanitarian settings, and will be used when available | Uptake increased | RIJ & PS. 27.11.2023 |
| 6 | Chi et al. 2015a | Burundi & Uganda | Other (Qualitative study) | Women of reproductive age | Maternal | SBA, ANC, CPR & Unmet need for FP | NA | Armed conflict | Factors affecting maternal health services utilization: financial barriers; level of household, community, and facility support; proximity; community perceptions | Conflict affects maternal health services utilization in diverse ways | Uptake remained unchanged | GCA, RIJ & PS. 27.11.2023 |

*(Continued)*

**Table 5.** (Continued)

| No | Author | Country | Study Design | Population | Types of HSB | Types of maternal health | Types of child health | Nature of armed conflicts | Result | Result Effects | Primary Outcomes | Name of extractors and dates |
|---|---|---|---|---|---|---|---|---|---|---|---|---|
| 7 | Chi et al. 2015b | Burundi & Uganda | Other (Qualitative study) | Women | Maternal | SBA, HIV/AIDS | NA | Armed conflict | Limited access to and poor quality of maternal and reproductive health services due to conflict. Abduction of health workers in Northern Uganda. Killing of health workers and ethnic bias in Burundi | Poor or no access to skilled services during the conflict, which has many perceived consequences | More than one outcome | GCA, RIJ & PS. 28.11.2023 |
| 8 | Chukwuma et al. 2019 | Nigeria | Cross-sectional | Women of reproductive age | Maternal | ANC, health facility birth & SBA | NA | Armed conflict | Reduction of ANC use by 13%, use of 4 ANC visits by 35%, deliveries in health facilities by 38%, and skilled birth attendance by 26% during conflict. | Boko Haram insurgency reduced the probability of any antenatal care visits, of delivery at a health center, and of delivery by a skilled health professional. | Uptake declined | RIJ & PS. 28.11.2023 |
| 9 | Chukwuma et al. 2021 | Kenya | Cross-sectional | Pregnant women | Maternal | ANC | NA | Civil conflict | When an initial ANC visit occurs in a facility within 10,000m of any conflict event, the client receives fewer recommended components of care. | The study demonstrated that conflict is an important determinant of non-utilization of ANC and the quality of care that women receive. | Other | RIJ & PS. 30.11.2023 |
| 10 | Druetz et al. 2022 | Burkina Faso | Quasi-Experimental | Pregnant women | Maternal | Caesarian section rate, assisted delivery and ANC | NA | Insurgency | 7.7% reduction of caesarean sections; 2.5% reduction of assisted deliveries; 1.8% reduction of ANC visits. | Insecurity level in a community is negatively associated with the use of maternal healthcare services. | Uptake declined | RIJ & PS. 30.11.2023 |
| 11 | McGinn et al. 2011 | Sudan; Uganda; DRC | Cross-sectional | Women of reproductive age | Maternal | Contraceptive use; knowledge of contraceptives | NA | Armed conflict | Low knowledge of contraceptive methods; Low use of modern contraceptive methods. | Low level of contraceptive use and knowledge associated with poverty, conflict, and low level of school attendance | Uptake declined | GCA, RIJ & PS. 04.12.2023 |

*(Continued)*

**Table 5.** (Continued)

| No | Author | Country | Study Design | Population | Types of HSB | Types of maternal health | Types of child health | Nature of armed conflicts | Result | Result Effects | Primary Outcomes | Name of extractors and dates |
|---|---|---|---|---|---|---|---|---|---|---|---|---|
| 12 | Namasivayam et al. 2017 | Uganda | Cross-sectional | Women of reproductive age | Maternal | Contraceptive use; ANC; prenatal care; SBA; Institutional delivery | NA | Civil conflict | Lower odds for contraception use and institutional delivery for women in the north of Uganda across time; Higher odds of SBA in Northern Uganda. | Conflict generally has a negative effect on healthcare utilization though some indicators show the opposite development: higher odds of SBA in the north; this may be due to targeted humanitarian interventions in the crisis region. | Uptake increased | RIJ & PS. 05.12.2023 |
| 13 | Nattabi et al. 2011 | Uganda | Cross-sectional | Patients; HIV Clinic Attendees | Maternal | Use of Family Planning Services | NA | Insurgency | 96% knew at least one FP method; 38% are currently using FP; low number of doctors in the region as a structural barrier to FP | Very low use of family planning methods despite high level of knowledge. | Other | RIJ & PS. 05.12.2023 |
| 14 | Ojeleke et al. 2022 | Nigeria | Cross-sectional | Children <5; women of reproductive age | Maternal and child | Childbirth at health facilities and ANC visits | vaccination | Insurgency | Exposure to conflict events significantly lower odds of completing 4 ANC visits, birth in a health facility and receive child immunization | conflict has a negative impact on healthcare utilization. | Uptake declined | RIJ & PS. 06.12.2023 |
| 15 | Sato, 2019 | Nigeria | Pre-Post | Children under 5 | Child | NA | Vaccination status | Insurgency | Living 10km from conflict significantly reduces odds the child will receive any vaccination (47.2%), BCG (45%) and DPT1 (48%) | Armed conflict has a negative effect on the uptake of childhood vaccinations; Maternal education and wealth are strong predictors. | Uptake declined | GCA, RIJ & PS. 08.12.2023 |

(Continued)

**Table 5.** (Continued)

| No | Author | Country | Study Design | Population | Types of HSB | Types of maternal health | Types of child health | Nature of armed conflicts | Result | Result Effects | Primary Outcomes | Name of extractors and dates |
|---|---|---|---|---|---|---|---|---|---|---|---|---|
| 16 | Sato, 2021 | Nigeria | Cross-sectional | Children under 5 | Child | NA | Vaccination status | Insurgency | Conflict events in Nigeria had weak or no effects on vaccination uptake while Boko Haram insurgency had a significant negative effect on vaccination | Boko Haram insurgency had a significant negative effect on vaccination; other forms of conflict that took place in Nigeria during this time had significantly weaker effects on vaccination uptake. | Uptake declined | RIJ & PS. 13.12.2023 |
| 17 | Sato et al. 2022 | Nigeria | Longitudinal | Children under and over 5 | Child | NA | Measles vaccination; measles infections | Insurgency | Regions affected by Boko Haram have the most measles cases and lowest vaccine coverage levels | Boko Haram insurgency led to the reduction of the vaccination uptake. | Uptake declined | RIJ & PS. 13.12.2023 |
| 18 | Solanke, 2018 | Nigeria | Cross-sectional | Mothers | Maternal | Use of maternal health services (specifically: ANCs and PNCs) | Not reported | Insurgency | Relevant individual characteristics: work status and education. Relevant community characteristics: community poverty concentration, community education | Community characteristics were more significant in predicting maternal healthcare use than individual characteristics. | More than one outcome | GCA, RIJ & PS. 15.12.2023 |
| 19 | Tyndall et al. 2021 | Nigeria | Cross-sectional | Children <5; Mothers | Maternal and child | ANC; Initiation of Early Breastfeeding | Measles Vaccination; Proportion of <5 severely underweight | Insurgency | Improvements of all major indicators over time; peak in 2016 when humanitarian aid was most prevalent; Borno state starts of all indicators very poorly in 2008 and then improves and even surpasses other states | Armed conflict may have a short-term positive effect on healthcare service usage as the conflict brings humanitarian action to regions where healthcare was underfunded and infrastructure insufficient before the conflict. | Uptake increased | RIJ & PS. 18.12.2023 |

*(Continued)*

**Table 5.** (Continued)

| No | Author | Country | Study Design | Population | Types of HSB | Types of maternal health | Types of child health | Nature of armed conflicts | Result | Result Effects | Primary Outcomes | Name of extractors and dates |
|---|---|---|---|---|---|---|---|---|---|---|---|---|
| 20 | Zhang et al. 2021 | DRC | Cross-sectional | Children <5; Mothers | Maternal and child | ANC, SBA, receiving PNC, early initiation of breastfeeding (EIB) | disease prevalence: diarrhea, fever, cough, and dyspnea | Civil conflict | Mothers with higher self-reported insecurity are more likely to seek maternal healthcare services and to have children who are sick with one of the four measured illnesses | Possible reasons are availability and accessibility of maternal health services and mothers' fear of pregnancy-related complications. | Uptake increased | RIJ & PS. 20.12.2023 |
| 21 | Ziegler et al. 2020 | DRC | Cross-sectional | Mothers | Maternal | first trimester ANC, 4 ANC visits; SBA; Place of delivery; combination | NA | Civil conflict | Odds of delivery in health facility and using an SBA are higher in women living in conflict zones while odds of meeting all WHO requirements, have four ANC visits and to have an ANC in the first trimester are higher | Conflict affected different indicators in different directions; NGO activity in conflict affected may be a vector through SBA utilization and hospital births are increased. | More than one outcome | GCA, RIJ & PS. 22.12.2023 |

In addition to the impact of armed violent conflict on maternal and child health behaviors through reduced service provision and utilization [62, 68, 71], maternal and child healthcare utilization has been found to be affected in a variety of other ways. Healthcare services and their quality were negatively affected by factors such as inadequate funding, shortage of healthcare personnel, and destruction of infrastructure [20, 68]. Poor quality of healthcare services and previous unpleasant experiences in health facilities, contributed to a lower demand for maternal healthcare usage [55, 60, 61]. Male partners' attitudes or their negative perceptions of healthcare services emerged as significant barriers to positive health-seeking behavior [55, 65, 67, 70]. Individual and community education, socio-economic status and well-being, and the act of watching TV were identified as predictors of health service uptake [70].

## Risk of bias and quality of evidence

The quality of the studies included in the systematic review was rated using GRADE, as recommended by the Cochrane Handbook for Systematic Reviews of Interventions [74, 75]. This involved categorizing the quality of evidence into four levels: high, moderate, low, or very low. For each of the included studies, the evidence-level was determined, and the risk of bias assessed. Specifically, we ascertained that outcomes were a direct result of the investigation and that the process was scientific and reproducible. We also considered bias due to selection, detection, attrition, and reporting, and assessed quality with regard to imprecision, inconsistency, and indirectness [44]. The review was guided by an independently developed instrument based on the GRADE guidelines [46, 76], as shown in Fig 3.

The results of these assessments are summarized in the "quality of evidence" column of the Fig 3. All but five studies were considered to be of high quality. The studies of moderate quality showed evidence of detection, reporting, attrition biases and indirectness in quality [49, 63, 65, 68, 71].

| | | | Measurement of Quality of Evidence in Studies (GRADE) | | | | | | | | | |
|---|---|---|---|---|---|---|---|---|---|---|---|---|
| | | | Grading of Recommendations Assessment, Development and Evaluation (GRADE) | | | | | | | | | |
| | | | RISK OF BIAS (High/Low/Unclear) | | | | | | QUALITY ASSESSMENT - (High/Low/Unclear) | | | Final Judgement/Summary |
| No. | Author | Study Design | Selection | Detection | Reporting | Attrition | Publication | Summary | Imprecision | Inconsistency | Indirectness | Summary | Quality of Evidence (High, Moderate, low) |
| 1 | Ahmed et al, 2020 | Cross sectional | Low | Unclear | Low | High | Low | Unclear | Low | Low | Unclear | Good quality | Moderate |
| 2 | Babakura et al, 2021 | Other (Retrospective analysis) | Low | Low | Low | High | Low | Unclear | Low | Low | Low | Good quality | Moderate |
| 3 | Bayo et al, 2021 | Cross sectional | Low | Low | Low | Low | Low | Low | Low | Low | Low | Good quality | High |
| 4 | Casey et al, 2013 | Pre-post | Low | Low | Low | Low | Low | Low | Low | Low | Low | Good quality | High |
| 5 | Casey et al, 2017 | Pre-post | Low | Low | Low | Low | Low | Low | Low | Low | Low | Good quality | High |
| 6 | Chi et al, 2015a | Other (Descriptive qualitative | Low | Low | Low | Low | Low | Low | Low | Low | Low | Good quality | High |
| 7 | Chi et al, 2015b | Other (Descriptive qualitative | Low | Low | Low | Low | Low | Low | Low | Low | Low | Good quality | High |
| 8 | Chukuma et al, 2019 | Cross sectional | Low | Low | Low | Low | Low | Low | Low | Low | Low | Good quality | High |
| 9 | Chukuma et al, 2021 | Cross sectional | Low | Low | Low | Low | Low | Low | Low | Low | Low | Good quality | High |
| 10 | Druetz et al, 2022 | Quasi experimental | Low | Low | Low | Low | Low | Low | Low | Low | Low | Good quality | High |
| 11 | McGinn et al, 2011 | Cross sectional | Low | Unclear | Unclear | Low | Low | Unclear | Low | Low | Low | Good quality | Moderate |
| 12 | Namasivayam et al, 2017 | Cross sectional | Low | Low | Low | Low | Low | Low | Low | Low | Low | Good quality | High |
| 13 | Nattabi et al, 2011 | Cross sectional | Low | Unclear | Low | Low | Low | Low | Low | Low | Low | Good quality | Moderate |
| 14 | Ojeleke et al, 2022 | Cross sectional | Low | Low | Low | Low | Low | Low | Low | Low | Low | Good quality | High |
| 15 | Sato, 2019 | Cross sectional/ Pre Post | Low | Low | Low | Low | Low | Low | Low | Low | Low | Good quality | High |
| 16 | Sato, 2021 | Cross sectional | Low | Low | Low | Low | Low | Low | Low | Low | Low | Good quality | High |
| 17 | Sato et al, 2022 | Longitudinal | Low | Low | Low | Low | Low | Low | Low | Low | Low | Good quality | High |
| 18 | Solanke, 2018 | Cross sectional | Low | Low | Low | Low | Low | Low | Low | Low | Low | Good quality | High |
| 19 | Tyndall et al, 2021 | Cross sectional | Low | Unclear | Low | Unclear | Low | Low | Low | Low | Low | Good quality | Moderate |
| 20 | Zhang et al, 2021 | Cross sectional | Low | Low | Low | Low | Low | Low | Low | Low | Low | Good quality | High |
| 21 | Ziegler et al, 2020 | Cross sectional | Low | Low | Low | Low | Low | Low | Low | Low | Low | Good quality | High |

**Fig 3. Quality of evidence and risks of bias in included studies.**

## Discussion

A first notable observation is that all 21 included studies reported different outcomes for healthcare-seeking behaviors in SSA. In other words, there appears to be no shared standard as to what health outcomes to report in the context of armed violence. Second, while the majority of studies reported significant *decreases* in the use of health services for both maternal and child health, a few reported *increases* in the use of various health services, and a minority reported mixed results. That is, we do not find a consistent effect of violent conflict on healthcare-seeking behavior. Rather, the studies reveal a more complex picture. Besides, factors such as marital status, male partner attitudes, female education, income, media (TV viewing), education, humanitarian aid, type and intensity of armed conflict, and community poverty levels have been identified as determinants of health-seeking behavior and utilization in conflict and post-conflict settings in SSA. We first discuss findings related to healthcare-seeking behavior for maternal health services, and then transition to healthcare-seeking behaviors for child health services.

### Healthcare-seeking behavior for maternal health services

Three studies showed significant increases in healthcare-seeking behavior, primarily related to the uptake of family planning, antenatal care, skilled attendance at birth, and postnatal care services [51, 57, 69]. For example, Casey et al. [57] reported an increase in the use of family planning commodities from 7.1% to 22.6% in northern Uganda and from 3.1% to 5.9% in the DRC. Similarly, in the DRC, mothers who reported higher levels of insecurity were more likely to use maternal and child health services, including professional ANC services, skilled attendance at delivery, postnatal care, early initiation of breastfeeding, and early care seeking for common childhood illnesses [69]. The increase in maternal health utilization in situations of armed violent conflict may seem counterintuitive, but there are plausible reasons for its likelihood. Experiencing the carnage of war or loss associated with armed conflict could cultivate a greater appreciation for life (mother) and/or a heightened sense of protection for new life (infant pregnancy), thereby promoting positive health-seeking behaviors [1, 77, 78]. The human losses associated with armed conflict may also increase reproductive activity, whether out of a need to compensate for losses or for other reasons.

In contrast, about half (10) of the studies documented a significant decrease in healthcare-seeking behaviors for maternal health services. The most commonly affected health-seeking behaviors included the uptake of family planning, antenatal care, skilled attendance at birth, and health facility birth. Caesarean section rates and use of postnatal care services were also affected. Antenatal care attendance decreased by 27.9%, 13%, and 1.8% in South Sudan, Nigeria, and Burkina Faso, respectively [20, 60, 62]. Similarly, births in health facilities decreased by 6.9% and 38% in South Sudan and Nigeria, respectively, and skilled attendance at birth decreased by 26% and 2.5% in Nigeria and Burkina Faso, respectively [20, 62]. A cross-sectional survey in three African countries affected by armed conflict (Sudan, Uganda, and the Democratic Republic of the Congo) found a reduction in the use of family planning services of up to 16.2% [63]. Other studies reported overall reductions in family planning, antenatal care, postnatal care, caesarean section rates, and the quality of antenatal care provided to clients [61–65, 67].

Interestingly, four studies reported mixed outcomes for maternal health service use, even within the same study. For example, ANC attendance increased from 30.2% to 51.2% in Somali country but decreased from 30.1% to 27.9% in Puntland [49]. In northern Nigeria, conflict clusters are more likely to have four or more ANC visits than non-conflict clusters, while exposure to conflict reduces the likelihood of a woman giving birth in a health facility [66].

Similarly, exposure to conflict events also reduced child immunization uptake [66]. Another study in the same region of northern Nigeria (Borno state) found low levels of all maternal and child health indicators since the insurgency began. Over time, however, all indicators improved [68]. In the DRC, women living in high or extremely high conflict zones were more likely to deliver in a health facility or to have a skilled attendant at birth, but less likely to attend ANC than women living in moderate conflict zones [70]. This shows that the impacts of armed violent conflicts are context-specific, leading to underutilization of maternal health services in some areas and increased utilization in others. Differences in health-seeking behavior have also been observed before and during conflict. Interventions to address health inequities must therefore take into account the context and timing of the conflict.

## Healthcare-seeking behaviors for child health services

Similarly mixed outcomes were obtained with regard to healthcare-seeking behavior for child health services. Three studies reported that healthcare-seeking behavior for child health improved despite exposure to armed conflict events. The type of healthcare-seeking behaviors found to improve were vaccination (diphtheria-tetanus-pertussis (DPT), measles, and OPV), case management for pneumonia, and an overall increase in the use of health services [49, 69, 71].

Conversely, five included studies revealed an overall decline in healthcare-seeking behaviors for child health in the region. The observed declines included vaccination rates (BCG, DPT, and measles), the proportion of well-nourished children under five, and care-seeking for neonatal complications [53, 66, 68, 72, 73]. Specifically, in three states (Adamawa, Borno and Yobe) heavily affected by insurgency in northern Nigeria, having a child living within 10 km of a conflict zone was associated with a 47.2% lower likelihood of being immunized overall, 45% lower for BCG and 48% lower for DPT1 [53]. In the same states, the Boko Haram insurgency was found to have a substantial negative effect on vaccination, with the likelihood of a child ever getting vaccinated decreasing by 40% if born within the conflict areas [72]. Similarly, a geographical and time trend analysis of measles incidence and vaccination coverage in Borno and Yobe states (Nigeria) confirmed that exposure to violent activities by Boko Haram led to a reduction in vaccination uptake. The regions affected by insurgency exhibited the highest measles cases and lowest vaccine coverage levels [73].

Health seeking behavior often reflects the prevailing maternal and child health conditions, with these factors interacting in an interdependent manner to create a dynamic pattern of healthcare-seeking that remains flexible and susceptible to change [79]. The health system in SSA faces many challenges, and exposure to armed conflict is known to frequently disrupt health service delivery. As shown in this review, violent conflict also significantly affects health-seeking behavior, especially among the most vulnerable groups—women and children. Therefore, research efforts should examine how armed conflicts affect the demand for and supply of health services. In particular, it is important to examine how changes in health-seeking behavior during armed conflict contribute to overall maternal and child morbidity and mortality in SSA [4, 34].

## Factors influencing health-seeking behavior in armed conflict and post-conflict settings

During armed conflicts, maternal and child healthcare are affected by several factors, as illustrated in Fig 4. First, at the structural level, exposure to armed conflict negatively affects health service delivery and quality due to inadequate funding, shortages of health workers, and destruction of infrastructure [20, 62, 68, 71].

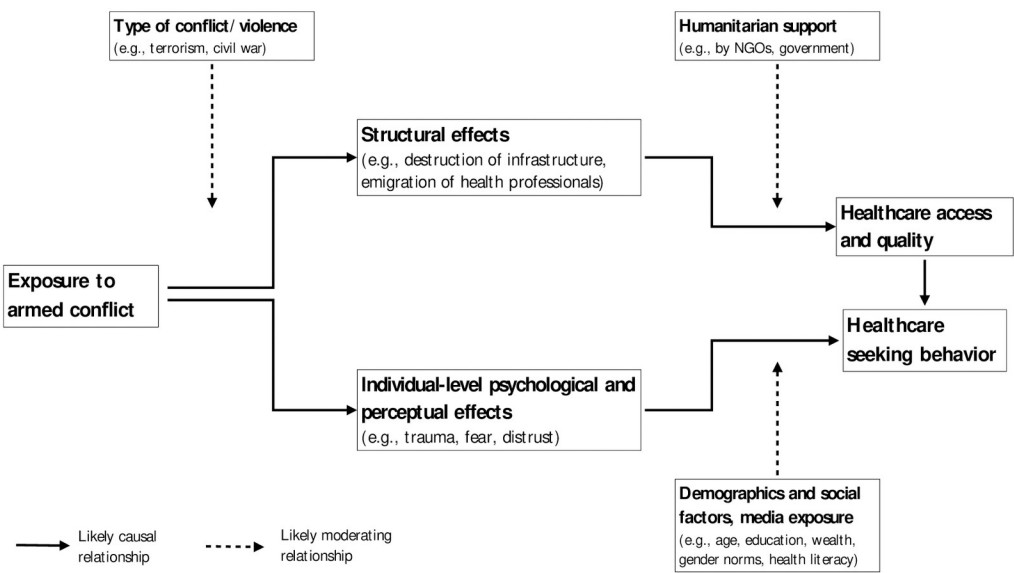

**Fig 4. Factors linking exposure to armed conflict to healthcare-seeking behavior.**

Second, alleviating these overall negative impacts, humanitarian interventions and aid deliveries were found to be an important catalyst for healthcare-seeking behavior. Seven of the included reviews highlighted the influx of humanitarian aid into armed conflict zones as a potential vehicle for improved maternal and child health utilization [49, 55, 64, 66, 68–70]. Non-governmental organizations (NGOs), both international and local, provide health services, thereby improving health service delivery. Some conflict-affected areas did not have good healthcare services before the conflict, making the influx of humanitarian aid an improvement to maternal healthcare services, thereby making them both available and accessible [68, 69]. The increase in healthcare services due to humanitarian intervention also had a positive effect on childhood vaccination. For instance, in Borno state (Nigeria), vaccination coverage rose from 12% in 2012 to 58% in 2016 [68]. It is important to note that these improvements resulting from humanitarian aid intervention are often not sustainable. When humanitarian aid withdraws at the end of a conflict, it is possible for the pre-war health status to return, in part due to donor dependency and the lack of a sustainability plan [49].

Apart from these factors influencing healthcare-seeking behavior by reducing access, a third relevant factor in maternal healthcare utilization that emerges is gender, as discussed in five of the examined papers [49, 55, 65, 67, 70]. The dominance of male decision-makers in healthcare choices, influenced by patriarchy or excessive masculinity, impacts family planning and household decisions. It also affects the uptake of ANC, PNC, or the use of SBA [49, 55, 65, 67, 70, 80]. Even in emergency caesarian-sections, seeking permission from male heads of household can have fatal consequences for both the mother and her newborn. Male opposition serves as a primary barrier to modern family planning, stemming from prejudice, safety concerns, or the desire for a larger family, especially after losing family members to the conflict [49]. Male partner opposition is also a significant barrier to HIV counselling and testing [55, 65].

Being married is negatively associated with ANC and PNC attendance and with the use of a skilled birth attendant [67, 70]. This effect is attenuated when the partner has a higher level of education or when the household's health decision-making power is not male-dominated. In

patriarchal settings, women also tend to receive less favorable treatment, experience delays, or not receive care at all if they are not accompanied to appointments by their husbands [55]. These situations are complicated during armed conflicts where husbands are deployed to defend the community or prosecute a war. Masculinity, as a significant determinant of health-care-seeking behavior in SSA and an emerging area of global health research, have been strongly linked to child health outcomes [80–82]. However, this review shows that its effects extend beyond child health to include maternal health and possibly other health-seeking behavior within households in SSA. Further empirical research is needed on this topic, with a particular focus on maternal health, to guide appropriate behavior-change interventions.

A fourth set of factors influencing healthcare-seeking behavior in conflict settings is related to individual attitudes of those seeking health, such as heightened perceived health risks among women. Two of the studies showed that women in conflict areas were more likely to deliver in hospitals, to seek early ANC services, and to make more ANC visits or use an SBA [64, 70]. In other words, many expectant mothers in armed conflict zones seek healthcare services, and this effect appears to be due to the perceived and increased risks of unsupervised delivery [69, 70]. Conversely, prior experiences of poor-quality healthcare services and unpleasant caregiver experiences (including the attitudes of healthcare workers) in health facilities in conflict-affected areas may negatively affect healthcare-seeking behavior [55, 60, 61].

Finally, in terms of demographic and community factors influencing healthcare-seeking behavior in areas exposed to armed conflict, women's level of education and household income were positive predictors of healthcare service uptake, alongside media exposure (watching television), which served as a means of providing awareness and general health promotion [67, 70]. At the community level, education had a positive and poverty a negative effect on indicators of healthcare utilization [67].

## Implication of the experience of violent conflicts on maternal and child health outcomes

Armed violent conflict has mixed effects on maternal healthcare utilization in SSA. Several contextual factors contribute to the direction of these effects, including gender norms, service quality, safety, access to health facilities, and fear of complications during pregnancy. More studies are needed to explore these factors further. Similarly, armed conflict in sub-Saharan Africa has been found to have mostly negative but mixed effects on children's health-seeking behavior. Studies have shown that several indicators of child health and healthcare utilization deteriorate during protracted conflict [66]. However, some studies also show an upward trend, such as an increase in health services due to humanitarian assistance in conflict regions, especially in regions where services were not extensive before the conflict [71]. Overall, it can be concluded that armed conflict has a negative impact on child health. However, these findings suggest that targeted health interventions can reach children in conflict-affected areas and even improve their access.

## Relationship between armed violent conflict and vaccination uptake

Of the six papers discussing childhood immunization, four concluded that armed conflict decreases vaccination uptake, one concluded that it increases vaccine uptake through increased availability owing to factors identified previously (humanitarian intervention), and one did not discuss the relationship. The overall negative impact of violent armed conflict on childhood immunization is clear, but this relationship is influenced by several factors. Security concerns and inaccessibility are important factors [71]. The Boko Haram insurgency reduced the odds of childhood vaccination by 35%, moderated by maternal education and household

income [53, 72]. Violent armed conflict had little effect on childhood vaccination uptake when the mother had some education, but it reduced the odds of vaccination by 64.3% when the mother was uneducated [72]. Importantly, income moderated the relationship in opposing ways. While armed conflicts did not affect the odds of vaccination for children from a poor household, those growing up in a non-poor household were 41.8% less likely to get vaccinated when exposed to armed violent conflict. This may be due to the prevalence of the Boko Haram insurgency in wealthier or urban areas compared to rural areas.

## General implication of armed violent conflict for population health

Armed conflict in SSA has a significant impact on maternal and child health-seeking behavior. The study has shown that armed conflict leads to reduced health service coverage and utilization, affecting reproductive, maternal, newborn, and child health interventions, nutritional status, and even child mortality. The escalation of armed conflict in the region undermines public infrastructure, including health systems, making it challenging to maintain adequate healthcare services. However, we have also seen that armed conflict can have unintended positive effects on maternal health-seeking behavior in regions with poor health infrastructure, especially when humanitarian health assistance is provided [64, 66, 68–70]. This highlights the complex and multifaceted influence of armed conflict on healthcare-seeking behaviors for maternal and child health in sub-Saharan Africa, and underscores the need for targeted community-based policy interventions and humanitarian assistance to mitigate the negative impact of armed conflict on health-seeking behavior in the region. There is also a clear need for further research to fill remaining gaps in our knowledge. A first area where research is needed is on the contextual determinants that shape the relationship between exposure to armed conflict and health-seeking behavior. While humanitarian aid seems to be one factor moderating the effect of armed conflict on healthcare-seeking behavior, it is unlikely to explain the full variation in findings. Another possible factor is the type of violence vulnerable populations are exposed to. Prior research has established that terrorism has strong negative effects on child health [35, 53], and it is noteworthy that some of the strongest reductions in healthcare-seeking behavior were in areas affected by attacks by Boko Haram, a known terrorist organization that explicitly and repeatedly has targeted civilians. It is possible that other forms of violence, such as insurgencies fought primarily between rebels and government forces, have less adverse effects [53]. These and other potential explanations for the mixed outcomes described in this review will need to be tested in future studies.

Another aspect that has so far received little attention is the effects of exposure to violent conflict on individual-level perceptions and psychological dispositions, including trauma, and their potential knock-on effects on healthcare-seeking behavior. Only six studies discuss subjective factors at all [19, 55, 60, 63, 65, 69], and none explicitly address perceptions of fear and psychological trauma. This is despite the fact that aspects such as increased fear of future violence and lost trust in state institutions have been hypothesized to drive many of the detrimental changes in healthcare-seeking behavior [34, 35]. Explicitly exploring these channels remains an important area for future research.

As with any review, this study has some limitations. The search strategy might have missed relevant studies due to the broad concept of healthcare-seeking behaviors for maternal and child health. In addition, including only published studies in English from SSA limits our ability to provide a global narrative on the influence of armed conflict on healthcare-seeking behaviors for maternal and child health. Furthermore, the absence of randomized controlled trials (RCTs) in our review is noteworthy, given that RCTs are considered the gold standard in

health research. This absence once again underscores the need for more high-quality research in this area to better inform evidence-based decision making.

## Conclusions

This review sheds light on the intricate and varied impact of armed conflict on healthcare-seeking behaviors for maternal and child health in SSA, highlighting the need for targeted policy interventions and increased humanitarian assistance to mitigate negative impacts. It serves as a valuable academic and policy resource for understanding population health behaviors in conflict and post-conflict contexts in SSA. In addition, it represents a first attempt to capture and contextualize the influence of exposure to armed conflict on healthcare-seeking behaviors for both maternal and child health services in the region. While previous reviews have focused primarily on the impact of armed conflict on infrastructure, health systems and service coverage, less attention has been paid to the impact of armed conflict on healthcare-seeking behavior. This is of concern, as timely healthcare-seeking is critical for effective management of existing health problems. Therefore, understanding the factors that shape healthcare-seeking behavior in the context of armed violence is critical to preventing harm to vulnerable populations, including children and mothers.

In addition to providing suggestions for further research, this review aims to help policy-makers and development practitioners or researchers fill the scientific knowledge gap on how exposure to violent armed conflict or post-conflict situations interacts with maternal and child healthcare-seeking behaviors in SSA. It also contributes to a deeper exploration and description of the micro-level factors that shape health-seeking behaviors in armed conflict settings within the subregion, through the lens of available evidence.

Armed violent conflict has mixed effects on child and maternal healthcare utilization in the region, indicating a need for further studies to elucidate the interrelated factors. While armed conflict disproportionately impacts child health compared to maternal health, it is noteworthy that exposure to armed violent conflict may have unintended positive effects in regions with poor health infrastructure, namely when humanitarian health assistance is unimpeded.

Policy intervention to change behavior for maternal and child health in the SSA need to be community-centered and integrated with humanitarian assistance to address the entrenched negative effects of their exposure. Knowledge or scientific evidence on this topic is still very limited in the region, and there is a clear need for further research to fill the gaps. For example, there is a vacuum on the contextual determinants that shape the relationship between exposure to armed violent conflict and healthcare-seeking behavior for maternal and child health. Also, it would be important to understand if health behavior is associated with the types of armed violence populations are exposed to, e.g., terrorism versus other forms of armed violence such as insurgencies, banditry, or organized crime. Similarly, it will be vital to better understand the impact of exposure to violent armed conflict on the perceptions, mental state, and mental health of individuals, including post-traumatic outcomes, and the relationship of these factors to healthcare-seeking behavior in the SSA region.

## Supporting information

**S1 Checklist. PRISMA-P (Preferred Reporting Items for Systematic review and Meta-Analysis Protocols) 2015 checklist: Recommended items to address in a systematic review protocol\*.**
(PDF)

**S1 File.**
(XLSX)

## Author Contributions

**Conceptualization:** Gbadebo Collins Adeyanju, Liliana Abreu, Max Schaub.

**Data curation:** Gbadebo Collins Adeyanju, Pia Schrage, Rabiu Ibrahim Jalo, Liliana Abreu, Max Schaub.

**Formal analysis:** Gbadebo Collins Adeyanju, Pia Schrage, Rabiu Ibrahim Jalo, Max Schaub.

**Funding acquisition:** Max Schaub.

**Investigation:** Gbadebo Collins Adeyanju, Pia Schrage, Rabiu Ibrahim Jalo, Liliana Abreu, Max Schaub.

**Methodology:** Gbadebo Collins Adeyanju, Pia Schrage, Rabiu Ibrahim Jalo, Liliana Abreu, Max Schaub.

**Writing – original draft:** Gbadebo Collins Adeyanju, Pia Schrage, Rabiu Ibrahim Jalo, Liliana Abreu, Max Schaub.

**Writing – review & editing:** Gbadebo Collins Adeyanju, Pia Schrage, Liliana Abreu, Max Schaub.

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
