## [Decision Letter · Decision Letter 0]

16 Jul 2024

PONE-D-24-12042Armed Violent Conflict and Healthcare-Seeking Behavior for Maternal and Child Health in Sub-Saharan Africa: A Systematic ReviewPLOS ONE

Dear Dr. Adeyanju,

Thank you for submitting your manuscript to PLOS ONE. After careful consideration, we feel that it has merit but does not fully meet PLOS ONE’s publication criteria as it currently stands. Therefore, we invite you to submit a revised version of the manuscript that addresses the points raised during the review process.

We look forward to receiving your revised manuscript.

Kind regards,

Adetayo Olorunlana, Ph.D.

Academic Editor

PLOS ONE

Journal Requirements:

"University of Hamburg provided funding as part of research support to its faculty members."

Reviewers' comments:

Reviewer's Responses to Questions

**Comments to the Author**

1. Is the manuscript technically sound, and do the data support the conclusions?

Reviewer #1: Yes

2. Has the statistical analysis been performed appropriately and rigorously? 

Reviewer #1: Yes

3. Have the authors made all data underlying the findings in their manuscript fully available?

Reviewer #1: Yes

4. Is the manuscript presented in an intelligible fashion and written in standard English?

Reviewer #1: No

5. Review Comments to the Author

Reviewer #1: Abstract:

Consider including a sentence on the study's practical implications.

Introduction:

Include recent studies to support the background information.

Explicitly link the rationale for the study to the existing literature.

Methods:

Provide specific participant selection criteria.

Discuss the reliability and validity of the instruments used.

Justify the choice of statistical methods.

Results:

Summarize key findings at the beginning of the results section.

Ensure all tables and figures are referenced in the text.

Discussion:

Focus more on the implications of the findings.

Discuss the impact of study limitations on the findings and future research.

Conclusion:

Include specific recommendations for future research or practice.

6. PLOS authors have the option to publish the peer review history of their article (what does this mean?). If published, this will include your full peer review and any attached files.

Reviewer #1: **Yes: **Sameer Kumar Jena

---

## [Author Response · Author response to Decision Letter 0]

20 Sep 2024

The point-by-point response to editor and reviewer's comments have been attached.

---

## [Editor Report · Decision Letter 1]

22 Dec 2024

Armed Violent Conflict and Healthcare-Seeking Behavior for Maternal and Child Health in Sub-Saharan Africa: A Systematic Review

PONE-D-24-12042R1

Dear Dr. Adeyanju,

We’re pleased to inform you that your manuscript has been judged scientifically suitable for publication and will be formally accepted for publication once it meets all outstanding technical requirements.

Kind regards,

Adetayo Olorunlana, Ph.D.

Academic Editor

PLOS ONE
---

## [Editor Report · Acceptance letter]

14 Jan 2025

PONE-D-24-12042R1 

PLOS ONE

Dear Dr. Adeyanju, 

I'm pleased to inform you that your manuscript has been deemed suitable for publication in PLOS ONE. Congratulations! Your manuscript is now being handed over to our production team.

Kind regards, 

on behalf of

Associate Professor Adetayo Olorunlana 

Academic Editor

PLOS ONE